# “A Light at the End of the Tunnel”—Post-COVID Condition and the Role of Rehabilitation and Recovery Intervention Delivered in a Football Club Community Trust: A Qualitative Study

**DOI:** 10.3390/healthcare13141733

**Published:** 2025-07-18

**Authors:** Steven Rimmer, Adam J. Herbert, Adam Leigh Kelly, Irfan Khawaja, Lewis A. Gough

**Affiliations:** Research for Human Performance and Health Laboratory, Centre for Life and Sport Sciences, Birmingham City University, Birmingham B5 5JU, UK

**Keywords:** post-COVID condition, rehabilitation, recovery, quality of life, exercise, football club community trust

## Abstract

**Background and Objectives**: This study explored the lived experiences of individuals with post-COVID condition (PCC) who participated in a 12-week exercise rehabilitation and recovery programme (PCCRRP) delivered by a professional football club community trust (FCCT). The aim was to understand the effects of the programme on physical function and quality of life (QoL). This study aims to address the gap in the literature of a lack of qualitative research exploring the experiences and perspectives of individuals engaging in exercise and physical activity as part of their recovery from PCC. Furthermore, it seeks to provide in-depth participant accounts to better understand outcome-level data. **Methods**: A qualitative approach was employed, involving semi-structured interviews with seven participants (mean age of 52 ± 8.54 years, with ages ranging from 45 to 60 years) following the 12-week PCCRRP to explore perceived changes in physical function and QoL. Thematic analysis was used to analyse the interview data, including participants’ narratives on their QoL experiences. **Results**: Participants reported improvements in exercise capacity, fatigue, and breathlessness, leading to enhanced physical function and QoL. They also experienced improvements in emotional well-being, including increased confidence and reduced anxiety. The programme’s focus on tailored exercise plans empowered participants to manage their symptoms and regain control over their lives. **Conclusions**: The PCCRRP delivered by an FCCT had positive effects on the physical function and QoL of individuals with PCC. This highlights the potential of FCCTs in providing effective rehabilitation and support for individuals with PCC.

## 1. Introduction

During the initial wave of the COVID-19 pandemic in 2020, it became evident that a significant number of individuals were experiencing persistent symptoms weeks and months after their initial Severe Acute Respiratory Syndrome Coronavirus 2 (SARS-CoV-2) cases [1]. This condition, known as post-acute sequelae of SARS-CoV-2 (PASC), long COVID, or post-COVID condition (PCC) [2], can affect individuals of all ages and across the spectrum of acute COVID-19 severity [1]. The projected increase in PCC cases led the Royal College of General Practitioners (RCGP) to call for a swift review of general practitioner (GP) practice provisions to effectively manage this influx of patients [3]. By the end of 2021, the Institute for Health Metrics and Evaluation [2] estimated that 3.92 billion individuals had contracted SARS-CoV-2, with an estimated 144.7 million developing PCC and 22 million experiencing persistent symptoms 12 months after the initial case [2].

PCC is recognised as a multisystem disease characterised by a wide range of symptoms, including but not limited to fatigue, post-exertional malaise, dyspnoea, cognitive impairment, headache, and musculoskeletal pain [4,5]. The most notable consequence of PCC is a diminished quality of life (QoL) and functional capacity, stemming from heightened sensitivity to physical, emotional, orthostatic, and cognitive stressors. These stressors trigger and intensify cyclical symptoms, hindering engagement in daily routines such as work, social interaction, and family responsibilities [6]. Given the increasing body of evidence highlighting shared symptomatology between PCC and other chronic conditions like fibromyalgia and myalgic encephalomyelitis/chronic fatigue syndrome (ME/CFS) [7], condition-specific rehabilitation may not be the most efficient use of scarce resources [8]. Nevertheless, various PCC rehabilitation approaches have been developed [9]. The challenge lies in tailoring community rehabilitation services to effectively address the unique needs of each individual with PCC, considering the diverse physical, psychological, and vocational manifestations of PCC, while maximising scarce resources.

In recent years, there has been growing interest in utilising professional football clubs to enhance health and well-being outcomes within their local communities [10,11,12,13]. football club community trusts (FCCTs) are the community arms and registered charities of professional football clubs. They are supported by local partners and are linked to their parent football clubs, offering programmes that package interventions aimed at improving physical activity (PA), reducing cardiovascular disease (CVD) risk factors, mental well-being, social interaction [12], and cancer rehabilitation exercise [13]. Cardiovascular disease, metabolic syndrome, and inactivity-related conditions present not only massive personal harm to individual sufferers and their families due to loss of functionality and livelihood, as well as pain, but also a significant financial burden for UK health care services [14]. As healthcare systems continue to grapple with the long-term effects of the pandemic, there is growing interest in the potential of exercise and PA as part of a rehabilitation strategy for individuals with PCC [15,16]. In June 2021, the Burton Albion Community Trust (BACT), the community arm of Burton Albion Football Club (BAFC), began the pilot delivery of a post-COVID condition rehabilitation and recovery programme (PCCRRP) to help individuals experiencing PCC. This initiative was part of a broader pilot project in Staffordshire funded by the National Health Service (NHS) to address the ongoing challenges of PCC and support those affected by the condition, as well as alleviate the pressure on an already overstretched NHS.

To provide a rigorous lens for understanding the complex recovery process and the mechanisms through which the PCCRRP fostered positive change, this study was conceptually guided by self-determination theory (SDT). Developed by Deci and Ryan [17], SDT posits that individuals possess three innate psychological needs—autonomy, competence, and relatedness—whose satisfaction is fundamental for intrinsic motivation, well-being, and sustained behavioral change. In the context of rehabilitation, autonomy refers to feeling a sense of control and choice over one’s recovery journey and interventions. Competence relates to feeling effective and capable in managing symptoms and performing exercises, and relatedness involves experiencing a sense of belonging and connection with others. This framework offers a valuable perspective for interpreting how a personalised, community-based intervention might enhance participant engagement and facilitate improved physical and mental well-being in individuals with PCC.

While emerging evidence suggests potential benefits of exercise in managing PCC symptoms [15,17,18,19], there is a lack of qualitative research exploring the experiences and perspectives of individuals engaging in exercise and PA as part of their recovery. Given the success of FCCTs in promoting health and well-being in various populations [10,11,12,13,20], it is crucial to explore their potential in addressing the needs of individuals with PCC. Understanding the lived experiences of individuals suffering with PCC is crucial for developing effective and patient-centred rehabilitation and recovery programmes. Therefore, the aim of this study is to explore the lived experience of individuals who participate in an exercise rehabilitation and recovery programme delivered by an FCCT and its effects on the physical function and QoL of individuals with PCC. Moreover, this study aims to address the gap in the literature regarding a lack of qualitative research exploring the experiences and perspectives of individuals engaging in exercise and PA as part of their recovery from PCC. Furthermore, it seeks to provide in-depth participant accounts to better understand outcome-level data.

## 2. Materials and Methods

A qualitative approach was employed, involving semi-structured interviews with seven participants (mean age: 52 ± 8.54 years) following the 12-week PCCRRP, to explore perceived changes in physical function and quality of life (QoL). Thematic analysis was used to analyse the interview data, including participants’ narratives on their QoL experiences. All participants were adults aged 18 years or older who had undergone a medical assessment at the PCC clinic by their primary care provider (PCP). Participants were referred to the BACT by their PCPs to undertake a PCCRRP. Participants were then recruited from the BACT by Birmingham City University (BCU) to participate in this study. Football fan status (football fan, non-football fan, fan of the host club, or fan of another club) was collected to explore its potential influence on participant engagement and adherence within an FCCT setting. The PCCRRP utilised a 12-week personalised exercise referral scheme (ERS) delivered twice weekly within a community setting. The PCCRRP included supervised low-to-moderate-intensity exercise sessions consisting of a combination of aerobic, stability and mobility, and strength-based exercises. The programme’s components (aerobic, stability and mobility, and strength) are designed to address the diverse physical, psychological, and vocational manifestations of PCC, aligning with the understanding that PCC is a multisystem disease with a wide range of debilitating symptoms like fatigue, dyspnea, and cognitive impairment. Each session was structured with a warm-up, main body, and cool-down. Exercises were personalised based on discussions with the participants, considering their previous activities, preferences, and areas they wished to explore. Progression was tailored to each individual’s needs, ensuring they felt supported in managing their symptoms and gradually increasing their activity levels.

Inclusion criteria for the PCCRRP included a previous diagnosis of COVID-19 with a current negative test result, being age 18 or older, the ability to walk independently for at least 20 metres, and access to transportation to attend gym-based sessions. Exclusion criteria included active COVID-19 symptoms (positive test), already receiving community-based rehabilitation, a diagnosis of ME/CFS, or a formal diagnosis of post-traumatic stress disorder (PTSD), clinically significant anxiety, or depression. This exclusion was primarily a safety measure for this pilot programme, ensuring that participants’ needs aligned with the scope of the FCCT’s exercise-based intervention. Individuals with pre-existing or clinically significant mental health diagnoses were typically managed through separate, specialised primary care pathways, as the PCCRRP was not designed to be a primary mental health intervention. This approach allowed the pilot to focus on the programme’s impact on general PCC symptoms and physical function within a defined cohort.

Prior to providing informed consent, all participants received comprehensive information about the intervention, including its rationale, methodology, potential benefits, and risks. Participation was voluntary, and participants were informed they could decline or withdraw without consequence prior to data analysis. Ethical approval for this study was granted by the BCU Health, Education and Life Sciences Faculty Academic Ethics Committee (ID#10203).

### 2.1. Interview Procedures

Interviews were conducted immediately following the completion of the 12-week PCCRRP. The semi-structured interview guide was developed collaboratively by all authors (S.R., L.G., A.H., A.K., and I.K.) to explore the participants’ lived experiences, beliefs, and attitudes towards exercise, physical activity, and overall recovery from PCC. The first author (S.R.), who had maintained prolonged contact with the participants through baseline assessment and delivery of the PCCRRP, conducted the interviews. The first author (S.R.) served as both an intervention provider and interviewer, having maintained prolonged contact with participants throughout the PCCRRP. This dual role facilitated rapport and deep insight into participant experiences but also raised the potential for researcher bias. To mitigate bias, the research team engaged in regular researcher reflexivity sessions, where the first author critically examined their preconceptions and influence on the data collection and interpretation. Thematic analysis and interpretation were subject to rigorous peer debriefing, with themes and sub-themes reviewed and confirmed by four co-authors (L.G., A.K., A.H., and I.K.). This established an audit trail for analytical decisions. The use of semi-structured interview questions, developed by all of the authors (S.R., L.G., A.H., A.K., and I.K.), allowed for flexibility and encouraged participants to share their thoughts and experiences freely. Preliminary themes and interpretations were presented back to a subset of participants for member checking, ensuring the findings accurately reflected their lived experiences.

### 2.2. Data Analysis

Following the interviews, the recorded audio files were transcribed verbatim. Thematic analysis was employed, utilising a six-step process of familiarisation, coding, generating themes, reviewing themes, defining and naming themes, and writing them up [21]. For the interview data, after transcription and immersion in the transcripts until saturation, coding revealed intriguing features within the data. These features were subsequently grouped into coherent themes. Drawing upon the tenets of SDT, the analysis specifically sought to identify how the themes related to participants’ experiences of autonomy, competence, and relatedness within the PCCRRP, thus providing a theoretically informed interpretation of their recovery journey. The themes and sub-themes were further reviewed by L.G., A.K., A.H., and I.K. for confirmation and clarification, ensuring the accuracy and authenticity of the captured information. To mitigate potential bias, pilot questions were tested beforehand to refine the interview structure and ensure the relevance of the collected data.

## 3. Results

Table 1 presents a comprehensive summary of the demographic profile of the study participants. A mixed-methods approach was employed involving semi-structured interviews with seven participants, whose ages ranged from 45 to 60 years, with a mean age of 52 ± 8.54 years. These individuals partook in the PCCRRP, a 12-week personalised ERS delivered twice weekly within a community setting. Exercises were personalised based on discussions with the participants, considering their previous activities, preferences, and areas they wished to explore. This individualised approach, where participants felt they could “go at [their] own pace” and that the programme was “led by me and what I enjoy doing”, was crucial for fostering confidence and engagement. The PCCRRP specifically explored perceived changes in participants’ physical function and QoL. Demographically, the cohort comprised a majority of female participants (71.43%), with males accounting for 28.57%. Football fan status was collected to investigate its potential influence on participant engagement and adherence within the FCCT setting. Specifically, 42.9% identified as football fans, 57.1% were non-football fans, with none affiliated with the host club, and 42.9% supported another club. Participants consistently reported perceived enhancements in their QoL following the PCCRRP. This was evident through their narratives describing a return to previously challenging daily activities, improved social engagement, and a renewed sense of normalcy (Table 2). Key aspects included a reduction in anxiety and depressive symptoms, diminished “brain fog”, and an increase in overall confidence and self-esteem. For instance, one participant stated, “A lot of it was the mental side of it for me. I was having flashbacks from COVID-19 the first time, a lot of anxiety and panic attacks. Now I feel like I’m living a happy life, and I make an effort. I will make an effort to go out and it’s great, feeling better about yourself.” Another commented on renewed feelings of hope. “The doctor mentioned about what Burton Albion were doing and I just thought it would be really good to help me. It felt that there was a bit of a light at the end of the tunnel.” These perceived changes fostered a renewed sense of self-efficacy and control. The following overarching themes were generated from the data.

The qualitative interviews conducted with PCC patients revealed several key themes (Table 2) related to their experiences with the illness and participation in a PCCRRP. These themes include the physical and emotional impact of PCC, challenges with diagnosis and medical care, coping mechanisms and support systems, and the recovery and rehabilitation process.

## 4. Discussion

The rich qualitative data provided significant evidence that the participants experienced significant improvements in their daily functioning and overall well-being, as evidenced by their ability to resume work, social interaction, and family responsibilities, which were previously hindered by PCC symptoms. The PCCRRP delivered by an FCCT elicit perceived positive effects on the physical function and QoL of individuals with PCC. Participants reported improvements in exercise capacity, fatigue, and breathlessness. Interestingly, no participants reported experiencing a worsening of symptoms after physical exertion, limiting their ability to engage in activities and necessitating rest and recovery periods such as those previously reported [22]. Participants expressed experiencing improvements in emotional well-being and feeling more confident and less worried, which aligns with the findings of Gerlis et al. [23]. This is likely due to the design of the PCCRRP, which focussed on tailored exercise plans that empowered participants to manage their symptoms and regain control over their lives, ultimately enhancing their overall well-being. The design of the PCCRRP may therefore be useful for other FCCTs to design their rehabilitation programmes.

### 4.1. Physical and Emotional Impact of Post-COVID Condition

Consistent with the previous literature [9,17,19,23,24,25,26,27], participants reported experiencing a wide range of persistent and debilitating physical symptoms, including fatigue, cognitive dysfunction, muscle weakness, pain, and breathlessness, prior to participation in the PCCRRP. These symptoms, as expected, significantly impacted their ability to perform daily activities and work and maintain family and social roles. The present study contributes a unique perspective to the growing body of research on PCC rehabilitation by focusing specifically on the role of FCCTs as a novel delivery setting for such interventions. While a systematic review highlights various rehabilitation approaches for older adults globally, including exercise training and respiratory rehabilitation, our qualitative exploration delves into the lived experiences of participants within an FCCT-delivered programme, emphasizing the potential of non-traditional community-based models. Other qualitative research has broadly explored the lived experiences of individuals with long COVID, detailing diverse symptomology and daily challenges. For instance, Humphreys et al. [28] specifically examined the role of physical activity in the lived experience of long COVID, and Thomas et al. [22] provided a longitudinal account of patients’ daily struggles and emotional impacts. Conversely, Cooper et al. [9] investigated the perceptions of both patients and general practitioners regarding access to community rehabilitation, uncovering systemic barriers and a lack of understanding. Our work, by contrast, centers on a specific and innovative community intervention, rather than the broader landscape of patient experiences or general access issues and reports positive findings for a small community-based rehabilitation programme.

The diversity in outcomes may be well explained by Humphreys et al. [28], who noted that diverse physical and psychological symptoms were intertwined, and the absence of a direct rehabilitation intervention in some cited works might explain differing trajectories of symptom improvement compared with the current study. Indeed, the physical symptoms often led to a decline in functional ability, making everyday tasks like walking, housework, and even concentrating on simple activities challenging and exhausting, similar to what was observed by Daynes et al. [25]. Furthermore, the chronic and unpredictable nature of PCC led to a loss of personal identity and a sense of grief for their pre-illness selves. This experience of a fractured identity, where participants felt like they had lost their “pre-illness selves”, can be interpreted as a profound threat to the psychological need for competence. According to SDT, competence is the need to feel effective and capable. By robbing participants of their ability to work, engage in household duties, or maintain social roles, PCC directly undermined their sense of mastery over their own lives, contributing to feelings of grief and inadequacy. Often, comparing themselves to healthy individuals or their pre-illness state led to feelings of frustration, inadequacy, and a sense of being left behind as others moved on with their lives, as previously reported [9].

The methodological approach of this study, a qualitative exploration through semi-structured interviews with a small sample of seven participants, provides rich, in-depth insight into individual experiences of the FCCT-delivered programme. This contrasts with larger-scale quantitative studies, such as the service evaluation by Smith et al. [19], which assessed clinical outcomes in 601 participants undergoing a blended digital and community-based rehabilitation programme. While Smith et al. [19] demonstrated significant improvements in QoL across a broader patient group, our study offers nuanced, subjective accounts of the programme’s impact within a specific setting, offering “*why*” something works. Daynes et al. [25] also provided quantitative evidence of clinical improvements in a cohort study of 30 individuals, focusing on fatigue, breathlessness, and exercise capacity. Furthermore, the systematic review and meta-analysis by Zheng et al. [26] synthesised evidence from 11 randomised controlled trials to assess the effectiveness of rehabilitation specifically for older adults with long COVID, a distinct and vulnerable population. While other qualitative studies, including Gerlis et al. [23], explored patient experiences of rehabilitation generally, our focused qualitative approach on an FCCT provides specific insights into this emerging model of care that larger, broader studies may not capture.

The present study emphasises the perceived positive effects of the FCCT rehabilitation programme, offering “a light at the end of the tunnel” for participants. This focus on benefits is crucial for advocating for such community-based initiatives. In contrast, some earlier qualitative research, such as that by Thomas et al. [22] and Humphreys et al. [28], frequently highlighted the significant challenges faced by patients, including unhelpful healthcare responses, the episodic nature of symptoms, and emotional burdens. Cooper et al. [9] similarly focused on the barriers to accessing rehabilitation from both patient and GP perspectives. The timing of release of this research (2025) also positions it within a maturing understanding of PCC, moving beyond initial symptom identification (as seen in earlier 2021 studies) and towards evaluating specific community-led intervention models. This evolution reflects the increasing need for diverse and accessible rehabilitation pathways as the long-term impacts of PCC become more apparent.

The emotional and psychological burden of PCC was also substantial, with participants reporting depression, anxiety, panic attacks, guilt, and a loss of self-esteem. With the loss of autonomy, the unpredictable nature of the symptoms meant participants lost a sense of control over their own bodies and daily lives. The program’s focus on “tailored exercise plans” and allowing participants to “go at my own pace” was therefore critical, as it directly worked to restore this sense of autonomy. By providing a framework where participants could make choices and regain control over their physical activity, the intervention helped mitigate anxiety and empowered them to become active agents in their own recovery. Furthermore, as observed by Romanet et al. [15], these physical and psychological symptoms were often intertwined, with physical incapacitation exacerbating emotional distress and vice versa. These findings underscore the need for comprehensive interventions that address both the physical and psychological aspects of PCC recovery, such as the tailored advice and support for managing exercise delivered by the PCCRRP.

### 4.2. Diagnosis and Medical Care

Consistent with previous research [9,22], a recurring theme in the data was the delay in receiving a PCC diagnosis and limited guidance and support provided by healthcare professionals. The participants stated that existing services were not equipped to handle the complex and multifaceted nature of the condition, in line with previous research [9], and often felt that their concerns were not taken seriously due to the lack of understanding and specific treatment options for PCC, leading to frustration, self-blame, and isolation (in agreement with [29]). Furthermore, this aligns with the findings of Cooper et al. [9], who reported that GPs were often reluctant to diagnose PCC due to the absence of a definitive diagnostic test and limited treatment options. This reluctance stemmed from the complexities of the condition, with its wide range of symptoms and the challenge of differentiating it from other chronic conditions like CFS/ME and fibromyalgia [7]. Additionally, GPs also faced difficulties identifying appropriate referral services, as PCC did not neatly fit into existing rehabilitation pathways [22]. Consequently, patients often turned to online communities and self-management strategies, highlighting the need for better education and support for both patients and healthcare providers. These platforms offered a space for individuals to connect with others experiencing similar challenges, fostering a sense of community and shared understanding. However, Humphreys et al. [28] warned of the potential for misinformation and conflicting advice within these online spaces, emphasizing the importance of professional guidance and reliable information sources. Consequently, due to the lack of available medical support, some participants resorted to experimental treatments, which sometimes worsened their symptoms, as previously reported [22]. Whilst this was an issue at the time of the study, with new information being produced daily, it is hoped that this may not be such a significant issue in the future.

Whilst the present study did not specifically investigate the impact of the PCCRRP on the diagnostic process, it suggests that referral to the program was often a turning point for the participants. The PCCRRP offered a structured approach to managing symptoms, providing a sense of hope, validation and “a light at the end of the tunnel” for individuals who had previously felt dismissed or misunderstood. This highlights the potential for specialised PCC services to not only aid in recovery but also improve the diagnostic experience for patients.

### 4.3. Coping Mechanisms and Support

Participants in this study, similar to Smith et al. [19] and Gerlis et al., [23] found the rehabilitation programme itself to be a significant coping mechanism. The structured tailored routine, social interaction, and professional guidance fostered a sense of hope and progress, contributing to improved mental well-being and QoL. This patient-centered approach directly nurtured participants’ sense of autonomy and competence, allowing them to take ownership of their recovery and adapt strategies to their individual needs. This aligns with the broader literature on self-efficacy in rehabilitation, where tailored interventions enhance a patient’s belief in their ability to overcome challenges. The importance of peer support was also evident in this study, aligning with the findings of Cooper et al. [9] and Gerlis et al. [23]. Connecting with others who shared similar experiences validated participants’ struggles and reduced feelings of isolation. Interestingly, while some studies like that of Jimeno-Almazan et al. [30] explored structured exercise protocols, our findings placed a particular emphasis on the psychosocial validation derived from shared experiences, which was a primary driver for engagement beyond mere physical benefits. While family and friends played a crucial role in providing emotional and practical support, as reported in the current study, Humphreys et al. [28] highlighted the potential for online communities to serve as valuable coping mechanisms, further broadening the scope of support systems for individuals with PCC. The importance of peer support was also evident in this study, aligning with the findings of Cooper et al. [9] and Gerlis et al. [23], and participants were open to non-exercise programmes where sessions would just be socials instead (e.g., coffee mornings). This could be key for other programmes, as this could be more inclusive and lower in cost compared with exercise-based interventions.

### 4.4. Rehabilitation and Recovery

Although some reported negative outcomes, the positive impacts of rehabilitation programmes on both physical and mental well-being are well documented in the literature. The current study, consistent with the findings of Smith et al. [19] Jimeno-Almazan et al. [18,30] Gerlis et al. [23] and Daynes et al. [25], demonstrated perceived significant improvements in physical function and QoL following rehabilitation. These studies collectively highlight the importance of exercise in improving physical function and reducing fatigue in individuals with PCC. The structured routine and social interaction aspects of the PCCRRP echo the positive experiences reported by Smith et al. [19] and Gerlis et al. [23], emphasizing the value of group-based rehabilitation in fostering peer support, motivation, and adherence to the programme, ultimately contributing to reducing feelings of isolation and promoting a supportive environment for recovery and improved mental well-being. This further underscores the critical role of relatedness in supporting sustained engagement and holistic recovery within the PCCRRP. Importantly, our findings build upon existing evidence by uniquely demonstrating how an FCCT setting can effectively leverage its community appeal and existing infrastructure to deliver a personalised exercise programme, successfully fostering positive outcomes and reducing symptom exacerbation, an issue noted in other contexts. This highlights a valuable, scalable model for community-based rehabilitation. Although the small sample size precluded a detailed analysis of football fandom’s effect on recruitment and engagement, the role of the FCCT as the delivery setting is a crucial contextual factor. The power of the club’s brand and its position as a trusted entity within the community may create a uniquely welcoming and non-clinical environment. This setting can reduce the perceived stigma associated with rehabilitation and may appeal to individuals, including non-football fans, who might be hesitant to engage with traditional healthcare services. The FCCT environment itself, separate from an individual’s interest in the sport, can foster a sense of local identity and social connection, which are key components of the psychological need for relatedness outlined in self-determination theory. While our data showed a mix of fans and non-fans, exploring how the unique community-centric identity of an FCCT influences participant motivation and outcomes remains a key area for future research with larger, more diverse cohorts.

The PCCRRP yielded perceived positive effects on participants’ mental well-being, mitigating anxiety and depression, reducing brain fog, and fostering increased confidence and self-esteem. This is particularly noteworthy as mental health support for PCC remains a significant gap in healthcare. The programme’s emphasis on personalised exercise plans instilled a sense of accomplishment and empowerment, contributing to improved mood and overall mental health. This directly reflects the satisfaction of the needs for competence and autonomy as outlined by the SDT, leading to enhanced intrinsic motivation for physical activity and overall well-being. This empowerment aligns with self-management strategies found in the works of Humphreys et al. [28] and Smith et al. [19], indicating that equipping individuals with tools to manage symptoms and gradually increase activity is crucial for sustained recovery. The holistic benefits of the PCCRRP enabled participants to resume daily activities and re-establish meaningful connections with their loved ones, with the overwhelmingly positive experience serving as a catalyst for sustained engagement in exercise and the prioritisation of physical and mental health beyond the programme’s conclusion. This study, therefore, contributes a qualitative understanding of community-based interventions, particularly within an FCCT framework.

### 4.5. Physical Impact of Post-COVID Condition

The programme yielded significant improvements in participants’ physical capacity and energy levels. One participant, who initially struggled to the point of being “out of breath” from walking up stairs and found it “such an energy drain”, experienced a profound shift, reporting, “I enjoyed going and got into a routine of going and I wasn’t worn out afterwards. Eventually I was just going home, and I’d be fine”. Another individual’s continuous exhaustion, described as being “so exhausted” by the end of the day, notably lessened, leading to them feeling “less drained”. Similarly, a male participant who had “not felt well for six months” and “just did not feel right” gained “much more energy now”. For those experiencing severe mobility limitations, the programme was transformative. One male participant recounted, “I couldn’t walk that far at all”, but after the intervention, he joyfully reported being “back to walking the dog”, signifying a return to meaningful daily life. The ability to manage daily tasks also saw a remarkable increase. A female participant shared her previous inability to perform household chores, stating, “Getting dressed and doing housework just didn’t really happen. I think my husband probably picked up a lot of the day-to-day stuff, things like cleaning the house and preparing the meals. If I did anything I would try to take the dogs for a walk, but literally the next day I’d feel so tired my whole body just felt drained. I couldn’t do anything the next day.” Following the PCCRRP, she triumphantly declared, “Now I have returned to my regular duties!” This extended to recreational activities, with another participant happily noting, “Now I am back cooking meals. No more ready meals! Sometimes I cook three meals at the same time depending on work”.

### 4.6. Positive Shifts in Emotional Well-Being and Mental Health

Beyond physical improvements, the programme significantly enhanced participants’ emotional well-being and mental health. Initial struggles with emotional distress were common. One female participant described, “I just used to cry loads. I was really grouchy”. However, the programme marked a turning point, with her now feeling “like I’m living a happy life, and I make an effort. I will make an effort to go out and it’s great, feeling better about yourself”. The psychological aftermath of COVID-19, including “flashbacks… a lot of anxiety and panic attacks”, was directly addressed, with one participant reporting, “touch wood I’m free from panic attacks and anxiety now”. The programme instilled renewed hope, as a male participant stated, “The doctor mentioned about what Burton Albion were doing and I just thought it would be really good to help me. It felt that was a bit of a light at the end of the tunnel”. This renewed optimism translated into a more proactive approach to recovery. For those experiencing both physical and mental burdens, the recovery was holistic. “I was just exhausted all the time and struggling with the mental health side of it as well. Now I am much less fatigued and mentally I feel stronger”. Some participants also grappled with feelings of self-blame and shame related to their condition, with one female asking, “Why is it me? Why have I got post-COVID-condition and everyone else seems to have caught COVID and been fine. Why am I really struggling? Am I to blame? Is it because I’m carrying a bit of extra weight? Is it because I’m not looking after myself as well?” Following participation in the programme, this individual, alongside others, reported a significant shift in perspective, moving towards a renewed sense of self-acceptance and a more positive outlook on their recovery journey, indicating a marked improvement in their emotional well-being and reduced feelings of self-blame.

### 4.7. Coping Mechanisms and Support

The need for robust coping mechanisms and support systems was evident. One participant bravely shared the profound impact her condition had on her home life and relationships. She confided, “I didn’t get much support off my partner at first and I was really struggling. I know if I tried to hoover the house then that’ll be it. Then I’d be shattered for a couple of hours.” This honest reflection reveals the isolating reality of living with an illness whose severity is not always immediately understood by those closest to you.

### 4.8. Strengths and Limitations

A key strength of this study is its qualitative approach, providing rich, in-depth participant accounts of perceived QoL improvements, which are lacking in the literature. Equally, the programme’s focus on empowering participants to manage their symptoms and tailor activity levels through exercise aligns with the emphasis on self-management strategies found in other studies, promoting long-term health benefits. However, the absence of a quantitative QoL instrument is a limitation, making direct statistical comparison with other studies challenging.

Another key strength of this study is that its findings offer several concrete recommendations for designing and delivering effective community-based rehabilitation for individuals with PCC. For the programme design, the success of the PCCRRP was heavily linked to its personalised nature, where participants felt they could work at their own pace, that the programme was led by themselves, and they enjoyed participation. This suggests that future programmes should be designed with the principles of self-determination theory at their core, ensuring participants have a sense of autonomy by offering choice and flexibility. Furthermore, the FCCT setting was crucial in making participants feel at ease, fostering a sense of community that helps build relatedness. This highlights the potential benefits of moving away from intimidating clinical settings and towards trusted community hubs that reduce stigma and encourage peer support.

From a clinical practice perspective, this study underscores a critical gap between diagnosis and care. Participants often felt lost, stating their GPs did not give them any advice, whereas the referral to the FCCT programme was described as helpful. This contrast points to a clear need for GPs and PCC clinics to actively identify and establish formal referral pathways to high-quality, community-based programmes to provide patients with accessible rehabilitation.

Finally, for policy and commissioning, this study provides evidence that community-led initiatives can alleviate pressure on the NHS while delivering effective holistic care. The findings support the exploration of funding models that formally integrate trusted community organisations like FCCTs into the broader PCC care strategy. This approach leverages existing community infrastructure and provides a scalable model for addressing both the physical and psychosocial dimensions of recovery.

We acknowledge the need for further investigation with a larger sample size. While demographic data was collected, the limited sample size prevented in-depth subgroup analysis by gender, ethnicity, or occupation. Future research with larger and more diverse cohorts could explore how these demographic factors influence experiences and outcomes in PCC rehabilitation. While football fan status was collected to assess its potential role in programme engagement within an FCCT setting, the small sample size and homogeneity of the cohort (e.g., all white British) precluded meaningful statistical or thematic analysis of this variable. Absence of a control group makes it difficult to definitively attribute all observed benefits solely to the programme, as other factors like social interaction and support from specialists could also contribute to improvements.

### 4.9. Future Recommendations

Future research should prioritise the development of standardised outcome measures, encompassing a wider range of physical and psychological parameters, including exercise capacity, cognitive function, and healthcare utilisation. This will facilitate a more robust comparison and evaluation of different rehabilitation approaches, ultimately leading to the development of optimal, evidence-based interventions for individuals suffering with PCC.

## 5. Conclusions

The present study concludes that the use of a PCCRRP generally improved participants’ physical and mental well-being. Based on the qualitative data in this study, this is likely due to the individualised exercise programme providing a sense of autonomy and the support provided throughout the programme from the community setting (i.e., BACT.). This study can offer some insight into future programmes to increase the likelihood of exercise-based interventions improving PCC. However, based on the limited sample size the scalability of this programme remains an important question that future research should seek to answer.

## Figures and Tables

**Table 1 healthcare-13-01733-t001:** Summary of demographic profile.

Variable	Total (%)
**Age** 52 ± 8.54 years	(Range: 45–60 years)
**Gender** Male Female	2 (28.57) 5 (71.43)
**Marital Status** Married or with Partner Single, Divorced, or Widow Other	5 (71.43) 2 (28.57)
**Ethnicity** White British	7 (100)
**Occupation** Care Education Manual Labour Unemployed	4 (57.1) 1 (14.3) 1 (14.3) 1 (14.3)
**Football Fan**	3 (42.9)
**Non-Football Fan**	4 (57.1)
**Fan of Host Club**	0 (0)
**Fan of Another club**	3 (42.9)

**Table 2 healthcare-13-01733-t002:** Themes, sub-themes, and participant quotes related to their experiences with PCCRRP.

Theme	Sub-Theme	Quote
Physical Impact of Post-COVID-Condition	From Debilitating Fatigue to Renewed Vigor: The Transformative Power of Rehabilitation	Prior to the programme, one participant struggled significantly, stating, “I couldn’t walk up the stairs without really being out of breath, and it was such an energy drain”. Following the PCCRRP, this participant reported a change. “I enjoyed going and got into a routine of going and I wasn’t worn out afterwards. Eventually I was just going home, and I’d be fine.” (Female 3)
	The Path to Recovery: Combating Post-COVID Fatigue and Restoring Daily Function	Before the programme, continuous exhaustion limited daily life, with one participant commenting, “Towards the end of the day I’d be so exhausted”. Following the programme, the participant expressed, “Now I feel less drained.” (Female 2)
	Renewing Inner Vitality: Overcoming Persistent Unwellness	One participant described, “I did not feel well for six months, I just did not feel right”. After the programme, a shift was reported, with the participant generally expressing, “I have much more energy now”. (Male 1)
	Reconnecting with Life: Regaining Mobility and Cherished Routines	Before joining the programme, one participant stated, “I couldn’t walk that far at all”, indicating a limitation in mobility that affected independence and QoL. Following the programme, the participant reported being “back to walking the dog”, which involved a return to a daily ritual and an increase in physical capacity, alongside reporting an improvement in quality of life. (Male 2)
	Reclaiming Daily Independence: From Household Burden to Active Participation	Participants consistently reported a notable increase in their ability to perform physical activities that were previously challenging or impossible due to PCC symptoms. This extended to daily tasks and general exercise capacity. “Getting dressed and doing housework just didn’t really happen. I think my husband probably picked up a lot of the day-to-day stuff, things like cleaning the house and preparing the meals. If I did anything I would try to take the dogs for a walk, but literally the next day I’d feel so tired my whole body just felt drained. I couldn’t do anything the next day”. Following the PCCRRP, this participant reported, “Now I have returned to my regular duties!” (Female 2)
Positive Shifts in Emotional Well-Being and Mental Health	Increased Confidence and Reduced Anxiety and Depression	Initial struggles with emotional distress were common, as one participant described, “I just used to cry loads. I was really grouchy. However, the program provided a turning point. Now I feel like I’m living a happy life, and I make an effort. I will make an effort to go out and it’s great, feeling better about yourself”. (Female 4)
	From Mental Anguish to Peace of Mind: Overcoming the Psychological Aftermath of COVID-19	“A lot of it was the mental side of it for me. I was having flashbacks from COVID-19 the first time, a lot of anxiety and panic attacks. Since the program I am free from panic attacks and my anxiety as all but gone”. (Female 3)
	Renewed Hope and Sense of Control	The initial referral itself instilled optimism, with one male participant stating, “The doctor mentioned about what Burton Albion were doing and I just thought it would be really good to help me. It felt that was a bit of a light at the end of the tunnel.” (Male 1)
	Holistic Recovery: Conquering Exhaustion and Fortifying Mental Well-Being	For those experiencing both physical and mental burdens, the recovery was holistic. “I was just exhausted all the time and struggling with the mental health side of it as well. Now I am much less fatigued and feel mentally stronger”. (Female 3)
	Navigating Self-Blame and Shame: The Psychological Weight of an Invisible Illness	Some participants also grappled with feelings of self-blame and shame related to their conditions, with one female asking, “Part of me was relieved that there was a diagnosis, but part was also really embarrassed as well. Why is it me? Why have I got post-COVID-condition and everyone else seems to have caught COVID and been fine. Why am I really struggling? Am I to blame? Is it because I’m carrying a bit of extra weight? Is it because I’m not looking after myself as well?” (Female 1)
Diagnosis and Medical Care	The Long Road to Recovery: From Delayed Diagnosis to Targeted Intervention	One participant reflected on the often-protracted journey to understanding her condition. She shared, “I was diagnosed with PCC about June 2021, a long time after initial infection”. This diagnosis, for this participant, marked the point before engaging with the FCCT programme as an intervention for her persistent symptoms. (Female 3)
	Bridging the Gap: From Diagnostic Hurdles to Essential Community Support	One participant recounted the initial struggles in seeking help. “Getting through to my GP was really difficult. Eventually I was referred to a PCC clinic and they were really good. They put me in touch with Burton Albion. Burton Albion was fantastic, really helpful”. (Female 2)
	Filling the Void: From Diagnostic Uncertainty to Empowered Recovery	One participant described her initial encounter with the healthcare system following her illness, stating, “The GP gave me the diagnosis of a PCC. He didn’t give me any advice, nothing at all! He’s a really great doctor and I’ve never heard anyone say anything bad about him but there wasn’t really any advice. I was skeptical about the program at Burton Albion at first, but thought I would try anything, and it really worked, I would recommend it”. (Female 3)
	Overcoming Fear and Finding Community: The Power of a Supportive Environment	For many participants, the journey to recovery began with a significant step and often a degree of trepidation. One woman openly shared her initial apprehension. “It bothered me to go into the gym the first time. I was scared, but then I was thinking positively as well, I just wanted to get better. I was put at ease very early by the instructors and soon realised there were other people just like me here”. (Female 1)
Coping Mechanisms and Support	The Invisible Burden: Navigating Unseen Illness and Unacknowledged Struggle within Relationships	One participant bravely shared the profound impact her condition had on her home life and relationships. She confided, “I didn’t get much support from my partner at first and I was really struggling. I know if I tried to hoover the house then that’ll be it. Then I’d be shattered for a couple of hours. This was the isolating reality of living with an illness whose severity isn’t always immediately understood by those closest to you”. (Female 4)
Recovery and Rehabilitation	Empowering Recovery: Pacing, Personalisation, and Building Confidence	One participant reflected on the programme’s personalised approach, saying, “I like the fact that I got to go at my own pace. I didn’t feel like I was being rushed. I like the fact as well that I was allowed time to build my own confidence in the gym. It was like, well, what do you feel comfortable doing today? What do you want to try? I like the fact that it was led by me and what I enjoy doing.” This patient-centered philosophy not only fostered physical progress but also empowered her to regain control over her rehabilitation. (Female)
	Positive Impact on Daily Life	Another participant’s experience demonstrated a return to daily routines and a significant uplift in personal well-being. She happily noted, “Now I am back cooking meals. No more ready meals! Sometimes I cook three meals at the same time depending on work.” (Female 1)

## Data Availability

The original contributions presented in this study are included in the article. Further inquiries can be directed to the corresponding author.

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
