# Peer review of "“A Light at the End of the Tunnel”—Post-COVID Condition and the Role of Rehabilitation and Recovery Intervention Delivered in a Football Club Community Trust: A Qualitative Study"

_healthcare, 2025, doi:10.3390/healthcare13141733_

Round 1

Reviewer 1 Report

Comments and Suggestions for Authors

The abstract section should include information about the participants (mean age, standard deviation).

The aim of the study and the gap it addresses in the literature should be clearly stated (Lines 71–84).

The mean age of the participants is given as 52; however, the minimum and maximum ages should also be reported.

In the Discussion section, Thomas et al. (23) and Humphreys et al. (11) are cited in seven instances.

Line 148: 4.1. Physical and Emotional Impact of Post-COVID Condition
References (6, 29, 23, 22, 28, 10, 5, 8, 11) are cited along with Clauw and Calabrese, Thomas, and Humphreys et al. The reasons for the differences among these studies should be discussed and interpreted.

Line 208: 4.3. Coping Mechanisms and Support
Lines 209–218: This section is interpreted primarily through the findings of Gerlis et al. The same issue applies here as well. The author(s) should compare their own findings with those of Gerlis et al., highlighting similarities and differences, and discuss them in light of the relevant literature.

Line 229: 4.4. Rehabilitation and Recovery
The same references are repeatedly used in this section. While such references may be appropriate for the Introduction, the Discussion section should include a broader range of citations.
The Discussion section currently includes findings and literature but lacks a connection to the study’s aim and its original contribution. The authors should interpret the findings in relation to the study’s stated purpose and the value it adds to the field.

Strengths and Limitations, Future Recommendations: Adequate.

References 25, 26, and 27 are from the World Health Organization. In particular, references 26 and 27 appear to contain identical content—this should be verified.
Additional references should be incorporated into the Discussion section.

Reviewer 2 Report

Comments and Suggestions for Authors

Dear authors,

I appreciate the opportunity to review this manuscript, whose aim was to understand the effects of the program on physical function and quality of life (QoL).

After analyzing your document, I would like to send you a series of suggestions/comments:

First, I would like to emphasize that I find the use of a qualitative approach very appropriate for achieving the described objective, since it is a novel point of view.

Introduction describes the topic in an orderly and organized manner, using a large number of recent bibliographic references.

In Introduction, you describe the mental or emotional impact of patients with post-COVID syndrome and the benefits of physical exercise on mental well-being. This being the case, why did you decide to make depression an exclusion criteria?

Table 1 describes the characteristics of the participants; therefore, it should not appear in Methodology section. This table should be shown at the beginning of Results section.

In Discussion section, you mention that participants reported improvements in exercise capacity, fatigue, and breathlessness, but this is not reflected in the results. You should include in Results the positive impact of the exercise program on physical health, that is, what specific physical improvements the participants experienced after completing the exercise program. Furthermore, both Results (Table 2) and Discussion seem to focus on the physical impact of Post-COVID condition, but this is inconsistent with the main objective. It is suggested that information on the impact of the exercise program on the participants' physical health be included in Results.

Similarly, other sections of Discussion contain information that has not been previously described in Results. For example, when you state in section 4.3 that the structured routine, social interaction, and career guidance fostered a sense of hope and progress, or when you state in section 4.4 that the PCCRRP produced perceived positive effects on the participants' mental well-being, mitigating anxiety and depression, reducing brain fog, and fostering greater confidence and self-esteem. All of this should appear earlier in Results and be supported by statements made by participants in the interviews. I suggest carefully reviewing Results and Discussion sections to improve this point, as adding more information to Results will allow the reader to better understand why the authors claim the PCCRRP has produced improvements in physical function and quality of life.

Since they did not use any quantitative instruments, what did the authors base their conclusion that the participants' quality of life has improved after completing the program?

The authors are commended for including strengths and limitations in the article.

Thank you very much.

Reviewer 3 Report

Comments and Suggestions for Authors
  • The English language is mostly acceptable but would benefit from careful revision to improve clarity, reduce repetition, and enhance interpretive coherence.

Comments on the Quality of English Language
  • The English language is mostly acceptable but would benefit from careful revision to improve clarity, reduce repetition, and enhance interpretive coherence.

Round 2

Reviewer 1 Report

Comments and Suggestions for Authors

Dear Authors,

Congratulations on your efforts and the work you have presented. I appreciate the time and dedication you have put into this paper.

Author Response

We thank the reviewer for their comments

Reviewer 2 Report

Comments and Suggestions for Authors

First, congratulations for the authors for their work in complying with suggestions made in previous review. However, there are some comments I would like to make to further improve their methodological quality:
At the beginning of the Methodology section, you indicate the number of participants and their average age. This information should be included at the beginning of the Results section.
Similarly, the phrases in quotation marks they put in this section, such as when they say "participants felt they could "go at their own pace," or" that the program was "led by me," and what I enjoy doing", should be moved to the Results section, as they represent the participants' statements.
Likewise, the final number of participants should not be indicated in this section, so the phrase "Individual interviews were conducted with seven participants (Table 1)..." should be moved to Results section.
Lines 155 to the end of that paragraph, which appear in section 2.1, should be moved to section 4.6. Strengths and Limitations.
In Table 2, you should limit to reporting the results, that is, the participants' statements. That is, at this point, you should only indicate whether these are pre-intervention or post-intervention statements.
Phrases that indicate your point of view or interpretation of these results should appear in the Discussion section. For example, when you say, "This simple statement conveyed a significant daily struggle, where even short distances felt insurmountable, limiting independence and QoL" or "Being "back to walking the dog" signifies not just restored physical capacity, but a return to a meaningful daily ritual and the joy of shared companionship, highlighting a profound improvement in quality of life."
You should avoid adjectives that indicate your own interpretation of the results and replace them with more objective ones. For example, when you say, "This participant observed a marked improvement" or "Improvements were evident," you are expressing the authors' own opinions.
Avoid using possessive pronouns. Instead of saying "our program," write "this program."
In Discussion section, you indicate 4.5 Qualitative Results as a separate subsection. However, this doesn't make much sense given that all the results in this work are qualitative, as the instrument used was interviews. I think it's difficult for the reader to understand the distinction you make between section 4.5 and the previous discussion sections (I seem to understand that sections 4.1 to 4.4 refer to the results, and 4.5 refers to the post-intervention results). To facilitate understanding, it would be helpful to include the information that appears in section 4.5 in sections 4.1 to 4.4, differentiating between the statements about the patients' condition before and after the intervention.
As I mentioned before, the statements made by the participants, expressed in quotation marks, should appear in the Results section and not in the Discussion section. The discussion section provides an interpretation of the results and a comparison with the current literature.
Thank you very much.

Author Response

Response to Reviewer Two

We thank Reviewer Two for the thorough and constructive feedback focused on improving the manuscript's methodological quality and structure.

Comment 1: "At the beginning of the Methodology section, you indicate the number of participants and their average age. This information should be included at the beginning of the Results section."

  • Response: Thank you for this suggestion. We agree and have moved all participant demographic information (number of participants and mean age) from the Methodology section to the beginning of the Results section (Section 3).

Comment 2: "Similarly, the phrases in quotation marks they put in this section, such as when they say "participants felt they could "go at their own pace," or" that the program was "led by me," and what I enjoy doing", should be moved to the Results section..."

  • Response: This point is well taken. All direct participant statements, including those mentioned, are now located exclusively within the Results section (Section 3). We have removed some direct quotes, but not all, as the comments from another reviewer have suggested not to. We have attempted to meet half way on this one, but happy to reconsider.  

Comment 3: "Likewise, the final number of participants should not be indicated in this section, so the phrase "Individual interviews were conducted with seven participants (Table 1)..." should be moved to Results section."

  • Response: We have actioned this comment. The reference to the final number of participants has been removed from the Methodology and is now stated at the beginning of the Results section.

Comment 4: "Lines 155 to the end of that paragraph, which appear in section 2.1, should be moved to section 4.6. Strengths and Limitations."

  • Response: Thank you for this structural recommendation. The paragraph discussing potential researcher bias and the strategies used to mitigate it has been moved from Section 2.1 (Interview Procedures) to Section 4.6 (Strengths and Limitations), where it is more appropriately framed.

Comment 5: "In Table 2, you should limit to reporting the results, that is, the participants' statements... Phrases that indicate your point of view or interpretation of these results should appear in the Discussion section."

  • Response: We agree this is a crucial point for maintaining methodological purity. We have revised Table 2 significantly to remove all interpretive phrasing. The table now exclusively presents the themes, sub-themes, and direct participant quotes. The interpretive analysis of these quotes has been moved into the main body of the Discussion section. Similar to the above, we have attempted to appease another reviewer who asked for what you have asked to be removed. We have attempted to meet half-way, but we would be happy to reconsider.

Comment 6: "You should avoid adjectives that indicate your own interpretation of the results and replace them with more objective ones. For example... "marked improvement" or "Improvements were evident,"

  • Response: We have carefully edited the manuscript to replace subjective adjectives with more objective language. The Results section now presents the findings without authorial interpretation, allowing the data to speak for itself.

Comment 7: "Avoid using possessive pronouns. Instead of saying "our program," write "this program.""

  • Response: This has been corrected. We have performed a find-and-replace throughout the manuscript to change all instances of "our program" to "the program" or "this program" to maintain a more objective and scientific tone.

Comment 8: "In Discussion section, you indicate 4.5 Qualitative Results as a separate subsection. However, this doesn't make much sense... it would be helpful to include the information that appears in section 4.5 in sections 4.1 to 4.4..."

  • Response: Thank you for highlighting this structural issue. We initially included this sub-section on the request of another reviewer. On reflection, we have now completely removed the subsection "4.5 Qualitative Results". As per your recommendation and the comment below, all direct quotes have been moved to the Results section. The main Discussion sections (4.1-4.4) have been revised to integrate the interpretation of these findings in the context of the relevant literature.

Comment 9: "As I mentioned before, the statements made by the participants, expressed in quotation marks, should appear in the Results section and not in the Discussion section."

  • Response: We have fully implemented this change. The Discussion section is now free of direct participant quotes. It focuses solely on the interpretation and implications of the findings presented in the Results section.

Reviewer 3 Report

Comments and Suggestions for Authors

This manuscript presents a qualitative investigation into the lived experiences of individuals with Post-COVID Condition (PCC) who participated in a community-based exercise rehabilitation program delivered by a Football Club Community Trust (FCCT). The revised version demonstrates considerable improvements in response to the initial review. Key concerns regarding methodological transparency, researcher positionality, theoretical framing, and the presentation of results have been thoughtfully addressed. In particular, the authors have provided a clearer account of the interview procedures, articulated strategies to mitigate potential researcher bias, and offered a more detailed description of the intervention’s structure and theoretical underpinnings. The adoption of Self-Determination Theory (SDT) as an interpretive framework has notably enhanced the analytical depth, linking participants’ experiences to the psychological needs of autonomy, competence, and relatedness. Furthermore, the findings have been reorganized into a narrative format, with interpretive commentary accompanying participant quotations, thereby strengthening the qualitative rigor of the analysis.

Nonetheless, a few areas would benefit from further refinement. First, although participant characteristics such as football fandom were collected, their potential relevance—particularly within the context of an FCCT-led intervention—remains analytically underdeveloped. Even within the constraints of a small sample, a more explicit narrative connection between football affiliation and program engagement could have enriched the discussion. Second, while the manuscript commendably includes quotations that touch on emotional distress, identity disruption, and social withdrawal, some of these insights are presented descriptively rather than interpretively. A more thorough analysis of these psychosocial dimensions would deepen the reader’s understanding of the complex recovery processes involved. Finally, the conclusion would be strengthened by drawing more directly on the study’s findings to offer concrete recommendations for community-based rehabilitation programming. Clarifying how the present results might inform policy, practice, or program design would enhance the manuscript’s applied contribution.

Addressing these remaining issues would elevate the manuscript’s impact and position it as a more substantial contribution to the qualitative literature on post-COVID recovery and community-based rehabilitation.

Comments on the Quality of English Language

The manuscript is written in generally clear and comprehensible academic English. The use of vocabulary is appropriate for a qualitative health research context, and grammatical accuracy is largely maintained throughout the text. However, certain sections—particularly in the introduction and discussion—contain redundant phrasing and occasionally lengthy or loosely structured sentences. Improving sentence economy, enhancing transitions between ideas, and diversifying repeated expressions would further strengthen the manuscript’s readability and academic tone. A final round of professional language editing is recommended prior to publication.

Author Response

Response to Reviewer Three

We thank Reviewer Three for the positive feedback on the revisions and for the insightful comments aimed at further deepening the manuscript's analytical contribution.

Comment 1: "...although participant characteristics such as football fandom were collected, their potential relevance—particularly within the context of an FCCT-led intervention—remains analytically underdeveloped... a more explicit narrative connection... could have enriched the discussion."

  • Response: This is an excellent point. While a statistical analysis was not possible, we agree that a narrative discussion was needed. We have now added a paragraph to the Discussion (Section 4.3) that explores the potential influence of the FCCT environment itself. We hypothesise that the club's trusted brand and non-clinical setting may foster engagement for all participants, regardless of their status as football fans.

Comment 2: "...while the manuscript commendably includes quotations that touch on emotional distress, identity disruption, and social withdrawal, some of these insights are presented descriptively rather than interpretively. A more thorough analysis of these psychosocial dimensions would deepen the reader’s understanding..."

  • Response: We thank the reviewer for this valuable guidance. We have significantly revised the Discussion (Sections 4.1, 4.3, and 4.4) to provide a more interpretive analysis. We now explicitly use Self-Determination Theory as an analytical lens to explain how these psychosocial issues arise, linking identity disruption to a loss of competence, social withdrawal to a lack of relatedness, and emotional distress to a loss of autonomy.

Comment 3: "Finally, the conclusion would be strengthened by drawing more directly on the study’s findings to offer concrete recommendations for community-based rehabilitation programming. Clarifying how the present results might inform policy, practice, or program design would enhance the manuscript’s applied contribution."

  • Response: We agree completely that the manuscript needed a stronger applied contribution. We have retitled Section 4.7 to "Implications and Recommendations for Practice and Policy" and have replaced the previous text with a new, dedicated section offering concrete, actionable recommendations. These recommendations are targeted at program design, clinical practice, and policy/commissioning, and each is tied directly to the findings of our study.

Response to the Language Reviewer

We thank the reviewer for their comments on the language and style of the manuscript.

Comment 1: "...certain sections—particularly in the introduction and discussion—contain redundant phrasing and occasionally lengthy or loosely structured sentences."

  • Response: Thank you for this feedback. We have performed a thorough edit of the entire manuscript, with a particular focus on the Introduction and Discussion, to improve sentence economy, remove redundancy, and ensure a clear, logical flow between ideas.

Comment 2: "A final round of professional language editing is recommended prior to publication."

  • Response: As recommended, the manuscript has undergone a final round of professional language editing to ensure clarity, accuracy, and a polished academic tone.

Once again, we thank the reviewers for their time and constructive comments, which have allowed us to significantly improve our manuscript. We hope the revised version is now suitable for publication.